# Security Risk Modeling in Smart Grid Critical Infrastructures in the Era of Big Data and Artificial Intelligence

**Abdellah Chehri** [1,*], **Issouf Fofana** [1] **and Xiaomin Yang** [2]

1   Department of Applied Sciences, University of Quebec in Chicoutimi (UQAC),
    Chicoutimi, QC G7H 2B1, Canada; ifofana@uqac.ca
2   College of Electronics and Information Engineering, Sichuan University, Chengdu 610064, China;
    arielyang@scu.edu.cn
*   Correspondence: achehri@uqac.ca

**Abstract:** Smart grids (SG) emerged as a response to the need to modernize the electricity grid. The current security tools are almost perfect when it comes to identifying and preventing known attacks in the smart grid. Still, unfortunately, they do not quite meet the requirements of advanced cybersecurity. Adequate protection against cyber threats requires a whole set of processes and tools. Therefore, a more flexible mechanism is needed to examine data sets holistically and detect otherwise unknown threats. This is possible with big modern data analyses based on deep learning, machine learning, and artificial intelligence. Machine learning, which can rely on adaptive baseline behavior models, effectively detects new, unknown attacks. Combined known and unknown data sets based on predictive analytics and machine intelligence will decisively change the security landscape. This paper identifies the trends, problems, and challenges of cybersecurity in smart grid critical infrastructures in big data and artificial intelligence. We present an overview of the SG with its architectures and functionalities and confirm how technology has configured the modern electricity grid. A qualitative risk assessment method is presented. The most significant contributions to the reliability, safety, and efficiency of the electrical network are described. We expose levels while proposing suitable security countermeasures. Finally, the smart grid's cybersecurity risk assessment methods for supervisory control and data acquisition are presented.

**Keywords:** smart grid; cybersecurity; machine learning; optimization; deep learning; cybersecurity risks; automated distribution network

## 1. Introduction

The concept of a "smart and sustainable city" is emerging with two flagship applications worldwide. The first target is to use better energy management—particularly with "smart" electricity grids promoting renewable energies. The second one is to deploy efficient mobility solutions to limit the automobile's use and, thus, limit greenhouse gas emissions. As useful as they are, information and communications technologies (ICT) are not an end in themselves.

A smart and sustainable city is an innovative urban strategy, using information and communications technologies to reduce the city's environmental footprint and improve citizens' quality of life. Indeed, the goal of using ICT is not only to increase the "IQ of the city" but to make it more sustainable and more pleasant to live in. This is a formidable challenge when we know that cities bring together an increasingly large population, expand and become denser with the attendant nuisances that this can imply [1].

The electricity sector is evolving towards a modern and automated distribution network. The demand for more digitized, connected, and integrated operations are growing in all sectors, so electricity companies must ensure a reliable power supply, with an approach based on efficiency and sustainable sources [2–4]. As the electrical grid merges and

becomes "smarter" with the resultant benefits of better connectivity, cybersecurity risks, and threats also increase.

Smart grid technology will allow a better adaptation to the dynamics of renewable energy and distributed generation, providing networks and consumers with more direct access to the benefits associated with these resources. An intelligent system's abilities will allow the easy and straightforward control of the bidirectional flow of electrical energy and facilitate the actions of monitoring, management, and support of resources at the distribution level.

Smart grids are autonomous and improve the effectiveness and efficiency of electrical power management, allowing utilities to optimize existing infrastructure, minimizing the construction of more power plants.

The main objective is to make the system more flexible to accommodate both the centralized renewable generation and all the generation and storage options linked to the distribution system [5–10].

For system security, it will bring about a radical change, both in supply and in the event of disasters, since the decentralization of generation will reduce the number of sensitive targets, such as large power plants.

From the environmental point of view, the modernization of the system will contribute much to the reduction in greenhouse gas emissions by promoting even greater distributed generation (especially concerning micro-generation through clean technologies), as well as the emergence of reliable sites for renewable sources, mainly hydro and solar, by avoiding the problems associated with intermittent supply and reducing the need to invest in a centralized fossil-source generation.

The analysis of threats in smart grid (SG) systems and the model of security threats in embedded systems helps to understand better attackers' weaknesses. For example, based on interactions in formalized incentive structures, the game theory approach allows us to carry out decision processes to address cybersecurity in monitoring and protection. Similarly, control from a coordinated cyber-attack perspective can improve security. In short, energy sector associations manage cybersecurity while maintaining critical power supply functions to ensure the modernized grid's reliability.

However, the most significant contributions to the reliability, safety, and efficiency of the electrical network have taken place in the development of intelligent optimization algorithms, such as genetic algorithms, neural networks, game theory strategies, reinforcement learning, vector support machines, among others. These previous strategies have made it possible to study the interactions in formalized security structures in response to demand in the energy markets. Consequently, modern SG control and monitoring systems have made rapid identification of critical infrastructure elements [11–17].

The International Organization for Standardization defines cybersecurity or cyberspace security as preserving confidentiality, integrity, and information availability in cyberspace. In turn, "cyberspace" is defined as "the complex environment resulting from the interaction of people, software and services on the Internet through technology devices and networks connected to it, which does not exist in any physical form".

In this work, we conduct a comprehensive overview and analysis of smart grid architecture and different security aspects in the era of big data and artificial intelligence. It is also a risk-based cybersecurity framework—a set of industry standards and best practices to help SG operators manage cybersecurity risks.

The paper's structure is as follows: Section 2 explores energy management in smart, sustainable cities. The main security threats in smart grids are given in Section 3. Section 4 provides the security-aware of SG infrastructures in the era of big data and artificial intelligence. A survey on risk modeling techniques is given in Section 5. We summarize the most efficient approach of mitigating cyber-attack risk on smart grid systems in Section 6. Section 7 concludes this survey paper.

## 2. Energy Management in Smart Sustainable Cities

The implementation of the smart and sustainable city, a complex system, requires new governance involving all the connected actors—local communities, companies, citizens—and a lot of research is required to draw its contours.

The concept of a "smart and sustainable city" is attractive. According to lifestyles and social and environmental issues, information and communication technologies to optimize and develop the city's functioning are indeed auspicious. Cities are implementing digital applications to give themselves a little more "intelligence" all over the world. That said, making a city more digital and smarter is not an end in itself. Information technologies are only one tool to achieve an objective: to make the city more pleasant to live in for its inhabitants, to make it cleaner, more economical, more fluid, and more participatory. In short, the challenge is to make the city more sustainable and livable, which, beyond technology, implies a new organization of its players, relying in particular on the participation of citizens.

The stakes are high. By 2025, around 58% of the world's population (4.6 billion people) will live in an urban area, and this rate will reach 80% for developed countries. By 2050, 75% of the world's population will live in cities, which are denser and more populated.

The challenge of urbanization is considerable: overpopulation, climate change, quality of the environment, access to energy, etc. Agglomerations consume around 65% of available primary energy and account for about 70% of greenhouse gas emissions, mainly due to the supply of energy for lighting, heating, and transport. To respond to these challenges, climate change, and deterioration in air quality, the city of tomorrow will have to structure itself.

Of all the possibilities that exist, energy management is the preferred application today by many cities. The energy issue is decisive, both for its effect on climate change and its impact on cities and citizens' bills. When it comes to energy, the smart city is often identified with the "smart grid". Thanks to smart meters equipped with sensors, it is possible to know the consumption of all buildings—housing, office buildings, etc.—particularly to identify the peak moments of energy consumption at the scale of a district and, ultimately, an entire city. These data make it possible to smooth consumption at peak hours by disconnecting devices and also to give consumers essential information to act on their behavior. This information on consumption, together with the decentralized production of electricity from renewable energies (wind, photovoltaic, cogeneration, geothermal energy, etc.) and electricity storage (mainly in batteries today), still allows for management of the production and use of electricity in an optimized way. Typically, the energy accumulated by photovoltaic panels placed on office buildings can be stored and delivered during the evening—that is, when offices are empty—to homes. Electric vehicles can be called upon to provide electricity during peak periods or serve as a storage system during off-peak hours [18–22].

## 3. Security Threats in Smart Grids

Smart grids reliability is based on the confidence, security, and availability of control of communication application systems [23].

Big Data processes an enormous number of datasets through computer devices and networks to generate useful information for supporting organizational decision-making. The architecture and framework of Big Data illustrate how hardware, software, networking, and data technologies orchestrate to perform the ultimate goal of this innovative methodology.

One of the sources of vulnerability resulting from integrating ICTs to SG is that all devices pass their data through the public network that is the Internet using the Internet Protocol (IP). However, this protocol has known weaknesses that can facilitate the risks of intrusions or interceptions of data. Yet, they have serious security gaps. Therefore, the safety in smart grids implies the protection and security of information.

The smart grid's major security requirements are the CIA triad (confidentiality, availability, and integrity). Before implementing cybersecurity measures and solutions that ensure safe and reliable operation, it is essential to understand the electrical network's safety objectives and requirements. The main goals and objectives are described below.

- Availability: guarantee access and timely use, and reliable information. Data availability is one of the most critical aspects of smart grids. A loss of availability represents the interruption of access and use of information, which could weaken the management and delivery of energy.
- Integrity: ensuring that information is not altered in a way unauthorized. This policy protects against modification and inappropriate destruction of data, ensuring this non-repudiation and its authenticity.
- Confidentiality: preserve the restriction of access and disclosure of the information. This policy addresses the protection of property of the data ensuring that sensitive data is not disclosed to unauthorized persons, entities, or processes [24].

Cybersecurity threats can be associated with the three major security requirements are discussed in Table 1.

**Table 1.** Malicious attacks on the smart grid.

| According to Threat | Security Objective Affected | Active or Passive | Examples |
|---|---|---|---|
| Interception (when personal unauthorized gets access to data, devices, or components cyber environment) | Confidentiality | Passive (usually cannot be detected but can be prevented with cryptography) | Denial of services (DoS), data traffic monitoring |
| Modification (when accessing) and modifications are made to data, environmental devices, or components cyber deliberately and illegally) | Integrity | Active (can be detected with cryptography) | Modification of control signals, modification of sensor data, modification of information (by example, energy use) |
| Interrupt (when data, devices, or components of the cyber environment are destroyed or turned to not available to delay, block, or impair the communication in the smart grid) | Availability | Active (can be detected, but usually not prevented) | Elimination of routing, software modification of deleting data, etc. |
| Manufacturing (when personnel not authorized inserts objects (for example, data or components) false in the system. | Authenticity | Active (can be detected with cryptography) | Saturation attacks, insertion of false control signals, insert of financial transactions bogus for-profit |

The National Institute of Standards and Technology (NIST) recommends individual security requirements specific to the smart grid, including cybersecurity and physical security [25].

As this article focuses on the security communication networks, below are some of the most critical cybersecurity requirements for intelligent electrical power systems based on the study developed in [26].

- Privacy: The smart meters and load management in networks intelligent electricity systems involve the use of patterns of electricity that could reveal private information [16,17,19,27]. For example, malicious users could use consumption patterns to determine how much energy is used in a residence or building and find out if consumers are or are not in them and thus be able to execute attacks. In addition, criminals could use information from these patterns to harm specific consumers. As a result, various privacy concerns must be addressed. Fortunately, the technologies related to privacy are very well developed, and the specific privacy solutions needed will depend on the type of protected communication resource [28].
- Attack detection and rapid response to incidents: The smart network electricity is a communication network that includes excellent coverage. Therefore, it is practically impossible to protect every node on the network. As a result, it is recommended to

perform profile checks, tests consistently, and make comparisons to monitor the state of network traffic to detect and identify abnormal incidents due to attacks [26].

- Continuity of operations: an information system of smart grids must have the ability to continue or resume operations in case of interruption of its normal functioning. The work presented in [26] introduces recommendations on policies and procedures of roles and responsibilities, storage centers alternative methods, alternative command and control methods, alternative control, recovery and reconstitution, and response to failure testing information regarding continuity of smart grid operations.

- Identification, authentication, and access control: The networks of smart electrical devices are made up of millions of devices electronic and intelligent information systems. Therefore, identification and authentication should be essential procedures for verifying a user or device's identity and a prerequisite to access resources in the smart grid's information system. This access control focuses on ensuring that resources are only accessed by staff appropriately and adequately identified. To achieve this, each node on the network must have essential cryptographic functions to perform authentications and data encryption [29].

- Audit and accountability: Periodic audits are used to detect gaps in security services to thoroughly examine smart grids' information system records [30,31]. Registration is required to detect anomalies; with the convergence of traditional electrical systems and information technology, the correct analysis of event information (for example, the power outage is necessary to understand what happened).

## 4. Security-Aware of SG Infrastructures in Era of Big Data and Artificial Intelligence

SG vulnerabilities are most common in smart meters, devices that interact with electricity supply and demand. This is a function of the geographic location where the meters are installed and the encryption level with which the energy consumption analysis algorithms are encoded [32,33].

Smart grids encompass the integration of information technologies for the electricity grid infrastructure. Consequently, the system's automatic operation allows effective options for both utility operators and clients—the preceding under the precept of guaranteeing the electricity supply's reliability and continuity.

Some supervisory control and data acquisition (SCADA) systems or elements were put in place dozens of years ago and are now impossible to update. Some of them were designed before well-founded cybersecurity principles were settled upon. SCADA system designers would claim that cybersecurity is not a concern since SCADA systems are not connected to the Internet. However, over time, SCADA systems began appearing on the Internet, and often with no cybersecurity. These systems must be replaced by more recent, safer equipment, but this is synonymous with significant investments and, therefore, often postponed.

On larger sites, the control system needs to be protected from attack within the SCADA network. Implementing an additional firewall between the corporate and SCADA network can achieve by imposing more restrictive rules. This will enable authorized service engineers to provide support and manage security, e.g., apply security mitigations, inspect log files, apply updates, etc.

Related studies in communications areas include communication network requirements for the main SG applications in domestic air networks, near air networks (NAN), and comprehensive air networks. For example, Bekara investigated security challenges in SGs based on Internet of Things (IoT). The author defined the primary security services that should be considered [34].

The concept has evolved, and today IoT encompasses many other technologies, including wireless sensor networks, machine-to-machine communications, and others, such as ZigBee, WiFi, NB-IoT, LTE, Bluetooth, among others. In Figure 1, it is possible to appreciate the myriad of information and telecommunications technologies that can operate in an electrical distribution system [35].

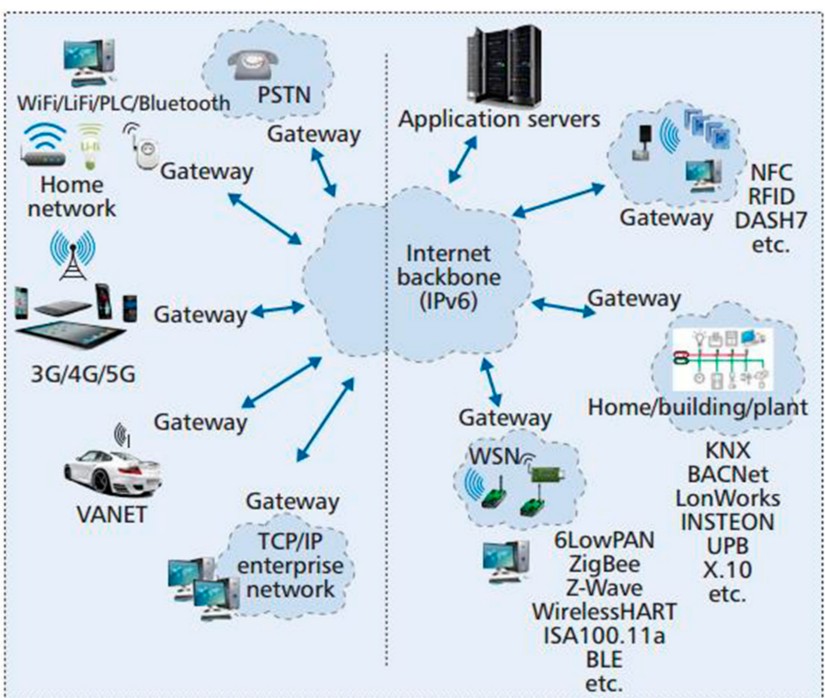

**Figure 1.** Myriad of information and telecommunications technologies [35].

### 4.1. The Enormous Potential of Big Data

The most important resource in the world is no longer crude oil, but data—according to *The Economist's* title from 6 May 2017. This lead story expresses the current assessment of big data well. Big data—a term for which there is no generally accepted definition—is pragmatic as a large amount of data, the analysis of which requires the use of tools that go beyond the classic application programs [36]. The acquisition, storage, analysis, maintenance, search, distribution, transmission, visualization, query, update, and data protection are challenges due to the database's size (as shown in Figure 2). There are three general approaches to analyze harmonized data across different sources: pooled data analysis, summary data meta-analysis, and federated data analysis [37–39].

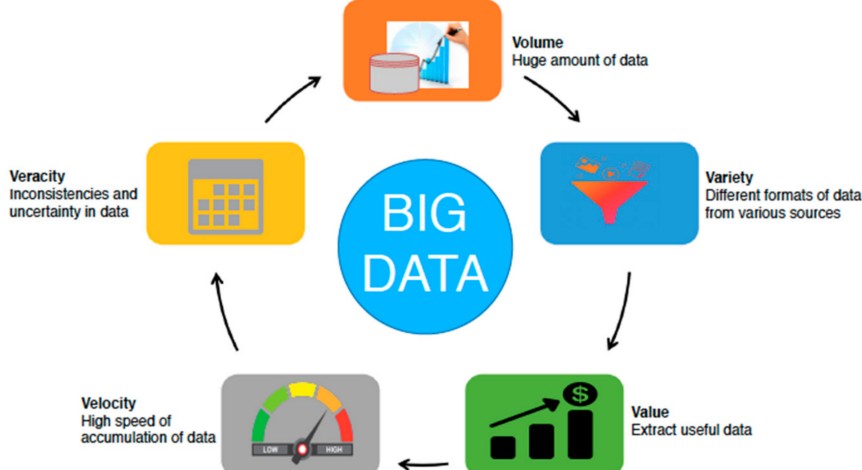

**Figure 2.** The properties of Big Data are reflected by 5Vs, which are veracity, volume, variety, value, velocity [40].

Thanks to forward-looking algorithms (i.e., prediction of consumption according to the weather, forecasting of production, etc.), the SG has a global vision in real-time or in advance of these energy offers and demands.

The smart grid's strength lies in using this data to automatically adjust the energy flows of the network to supply areas of energy need with electricity primarily from renewable sources.

Electricity distributors are now actively engaged in a double movement towards Big Data—the quantitative explosion of data digital available—and to Open Data—the update free disposal of this data in an open manner, which allows their reuse without technical restriction [40].

*4.2. Cybersecurity and Artificial Intelligence*

Cybersecurity is one of the many uses of artificial intelligence (AI) [41]. Buzzwords, such as machine learning, natural language processing, and robot-assisted process automation (RPA), are currently primarily associated with digitized production processes [41]. But these technologies have also long been used in cybersecurity. The spam filter, for example, is an excellent example of the application of machine learning that dates back to the early 2000s [42–44]. Of course, the methods have become more refined over time, and the systems now deliver analyses at a much higher level.

Today, the latest developments in artificial intelligence are already making a valuable contribution to improving digital security in the smart grid. The innovations in this area help to defend against a whole range of attack vectors. The five most common use cases are fraud detection, malware detection, intrusion detection, risk assessment, and user behavior analysis. Artificial intelligence is implemented more often than is generally known.

AI delivers insights that allow businesses to quickly understand threats, reducing response times and keeping businesses in compliance with security best practices. Artificial intelligence, 5G, and other technologies are poised to aid with these challenges, but the energy industry must continue to invest in getting ahead of cyberattacks [45]. Another AI application field is the detection and prevention of unauthorized access to network infrastructures (intrusion prevention), be it external or internal. Deep Learning (DL) systems also support user account monitoring. The AI algorithms examine user behavior and can thus detect anomalies—e.g., through different geolocations within a very short time, unusual working and access times, or the use of databases that were previously not or only rarely used [46–48].

On the other hand, machine learning (ML) helps to recognize patterns in data so that machines can learn from experience [49]. By leveraging cyber threat intelligence, smart grid users can respond to problems quickly and confidently [50,51].

The current security tools are almost perfect for identifying and preventing known attacks, but unfortunately, they do not quite meet the requirements of advanced cybersecurity. These solutions offer no protection against new, unknown attacks, zero-day attacks, and low and slow attacks. Therefore, a more flexible mechanism is needed to examine data sets holistically and detect otherwise unknown threats [52–69]. Machine learning, which can rely on adaptive baseline behavior models, is extremely effective in detecting new, unknown attacks: The combination of known and unknown data sets based on predictive analytics and machine intelligence will decisively change the security landscape [70–74]. Table 2 shows how AI can boost cybersecurity in SG.

**Table 2.** Artificial intelligence (AI) and Cybersecurity.

| How AI Can Help in Cybersecurity | References |
|:---:|:---:|
| Automated Detection | [52–56] |
| Quick Identification Errors | [57] |
| Secure Authentication | [58–60] |
| Faster Response Times | [61–64] |
| Cybersecurity without Errors | [65–68] |

## 5. Survey on Risk Modeling Techniques

To keep pace with these developments and not be helpless in the face of AI-based cyber-attacks, electric utility companies are ultimately almost forced to base their security strategy on similar technologies [75–78]. There are already many effective AI-based security solutions available, especially in endpoint protection. Unlike conventional signature-based protection technologies, these next-generation solutions focus on dynamic behavior analysis techniques and combine these with machine learning and intelligent automation [79–83]. Infections with malicious code are identified here based on their execution behavior within a few seconds and automatically blocked before damage can occur [83]. Machine learning capabilities ensure that the behavior analysis technology is constantly learning and, thanks to the constantly flowing information about threats, is continuously optimized [84].

Cybercriminals are still causing billions in damage using traditional attack methods, and without the use of artificial intelligence, it says a lot about the current state of IT security.

Some of the leading SG research technologies are mentioned in this section. The previous techniques are based on the dynamic integration of electrical engineering developments, energy storage, big data analysis, advances in information, communication technologies (ICT), wireless communication, and machine learning techniques [85–87]. Furthermore, advanced fault management is possible, thanks to the complete coordination of local automation. That is why these sophisticated systems can be used to protect essential consumers from interruptions. In this order of ideas, diagnostic techniques are essential in SG since they must be fault-tolerant [88,89].

### 5.1. CORAS Method for Security Risk Analysis

A literature review is used in this article to explore various security modeling techniques and their applicability in smart grid security [41]. The CORAS method for security risk analysis was used, as shown in Figure 3.

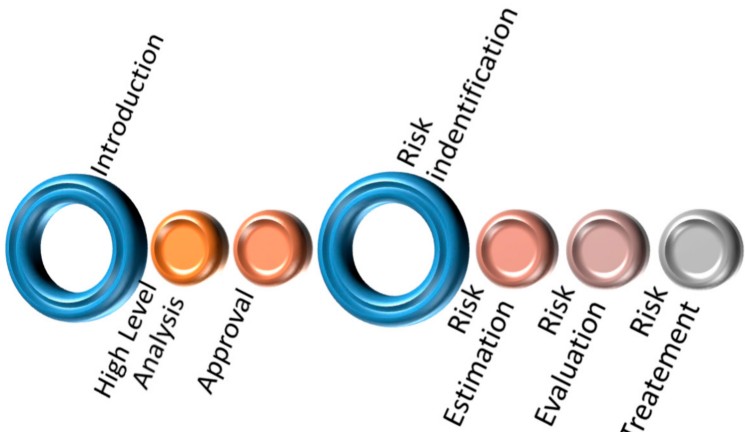

**Figure 3.** CORAS method for security risk analysis.

The electronic databases IEEEXplore and SpringerLink were used in this literature review. The work consists of a main qualitative study supplemented by a quantitative study. Building a database is not as easy as it sounds to create the comparative databases, and search keywords. Among these keywords, we can quote "attack tree security", "vulnerability analysis", "false data injection attack detection", "malicious behavior detection", "deep learning detection of electricity theft cyber-attacks", "fraud detection", "bow tie security", "anomaly detection method", "smart grids cyber-attack defense", and "CORAS security".

These data were sufficient to create a comparative database and apply high-level quality indicators. Table 3 shows the number of hits each of the keywords returned from IEEEXplore and SpringerLink databases.

**Table 3.** Classification based on security requirements.

| Attacks | References |
|---|---|
| Switching Attacks | [46–48,51,56,63,66–69,75,77,79,84,85,89–94] |
| DoS (Denial of Service) | [95–99] |
| Fraud Detection | [78,96,100] |
| Cyber Threat Detection | [14,29,96,97,101–104] |
| Data Integrity | [105–112] |
| Replay | [113–116] |
| Packet Dropping | [117–119] |
| Dynamic Load Altering Attack | [5,27,120–126] |
| Data Injection Attacks | [47,57,59,75,89,101,107,127–137] |
| Malicious Software (Malware) | [92,109,113,114,138–144] |
| Vulnerability Analysis | [80,104,145–147] |
| Anomaly Detection | [83,148–151] |

*5.2. Cyber Security Risk Assessment Methods for SCADA Systems*

This work has a qualitative approach. It intends to make a reflective analysis based on the documentary review on some methodologies implemented to evaluate cybersecurity risk applied to SCADA (supervisory control and data acquisition) systems for electricity companies. What are the appropriate methods to implement in electricity companies, taking into account vulnerabilities? What are the shortcomings and possibilities for improvement in the current plans?

- Method 1: Analysis, classification, and detection methods of attacks through wireless sensor networks in the smart grid and SCADA systems [152].
- Method 2: Detection of cyberattacks using temporal pattern recognition techniques [153].
- Method 3: A CPI-enabled firewall model for SCADA security in smart grid networks [154].
- Method 4: Combining ensemble methods and social media metrics to improve the accuracy of One Class Support Vector Machine (OCSVM) in intrusion detection in SCADA systems [155].
- Method 5: Unconditional security practical implementation for the IEC 60780-5-101 SCADA protocol [156].
- Method 6: SCADA approach as a service for the interoperability of micro-network platforms. According to [157], in the context of the development of smart grids, this work considered the interoperability of microgrid platforms. Various levels of interoperability were introduced with the respective requirements. The document's main objective was to propose a suitable hybrid cloud-based private SCADA architecture satisfying multiple needs within the interoperability of micro-network platforms while maintaining security constraint conditions. Interoperability between micro-networks will allow research institutions to exchange meaningful information, gain access to the pool of shared resources, and eventually, locally or remotely, borrow associated infrastructure for research activities.
- Method 7: Simulation platform for cybersecurity and critical infrastructure vulnerability analysis [158].
- Method 8: Pre-distribution key scheme with joint license support for SCADA systems [159].
- Method 9: Development of a secure and attack-resistant SCADA system using Wireless Sensor Network (WSN), Mobile Ad hoc NETwork (MANET), and the Internet [160].
- Method 10: Cascading dynamics vulnerability analysis in smart grids under load redistribution attacks [161].
- Method 11: Ensure operations in the industrial control system based on the SCADA-IoT platform using deep belief [162].
- Method 12: An improved algorithm based on optimization for intrusion detection in the SCADA network [163].

The list of the risk assessment methods described in this subsection is summarized in Table 4.

**Table 4.** Cyber security risk assessment methods for supervisory control and data acquisition (SCADA) systems.

| Method | References |
| --- | --- |
| Analysis, classification, and detection methods of attacks through wireless sensor networks | [152] |
| Detection of cyberattacks using temporal pattern recognition techniques | [153] |
| CPI-enabled firewall model for SCADA security in smart grid networks | [154] |
| Combining ensemble methods and social media metrics to improve the accuracy of OCSVM in intrusion detection in SCADA systems | [155] |
| Vulnerability Analysis | [156] |
| Data Integrity for cloud-based private SCADA architecture | [157] |
| Simulation and Malicious Software (Malware) | [158] |
| Replay and pre-distribution key scheme | [159] |
| Packet Dropping and attack-resistant SCADA system | [160] |
| Dynamic Load Altering Attack | [161] |
| Data Injection Attacks using using deep belief | [162] |
| Anomaly Detection and optimization for intrusion detection | [163] |

## 6. Mitigating the Risk of Cyber Attack on Smart Grid Systems

Protecting against today's cyber threats requires greater collaboration between engineers, IT managers, consumers, and security managers, who must share their knowledge to identify potential problems and attacks that affect their smart grid systems. Utilities need to consider how cybersecurity strategies will evolve. It is about staying current against known threats in a planned and iterative way. Having a good defense against cyber-attacks is an ongoing process and requires constant effort. Electricity companies must implement a complete program that integrates a good organization and adequate processes.

The traditional tiered approach to cybersecurity can only prevent and detect the less elaborate threats. In the meantime, modern cyber-attacks are carefully designed to bypass standard security controls by learning detection rules. In addition, traditional controls may not adequately counter insider threats, a form of insidious attack launched by those with legitimate access.

By leveraging AI and advanced big data analytics, cybersecurity technologies can generate predictive and actionable insights that will help you make better cybersecurity decisions and protect your smart grid against threats. They can also help the electric utility detect and counter threats faster by monitoring the cyber environment at speed and with a precision level that only machines can.

Artificial intelligence technologies are already integrated into tools, such as antivirus, EDR (endpoint detection and response) solutions, firewalls, data loss prevention, etc., that automatically respond to attacks by filtering malicious traffic. Vulnerability management has become a point of tension for operational teams due to the constant increase in the number of known vulnerabilities, difficulties in assessing the real risks induced, and prioritizing and automating patches' deployment. Indeed, of the thousands of vulnerabilities published each year, only a fraction is used by attackers. Besides, some systems are protected by perimeter defenses.

This complexity is driving vulnerability management tool vendors to integrate AI technologies into their solutions. The objective of AI applied to vulnerability management is to improve the discovery of active equipment, the scanning of vulnerabilities, the determination of associated risks connected with intelligence on the threat, the prioritization, and deployment of patches.

Establishing and maintaining a robust and adequately implemented cybersecurity awareness program for SG, several approaches (as shown in Figure 4) must be followed:

- Secured Remote Access: The mere protection by the combination of password and user name is by no means sufficient here. Encrypted connections, for example, via

VPN (virtual private network), are a better choice here [82]. These considerations already show that there is no universal security solution that fits all companies and electric utilities but that the corresponding measures must always be tailored to the operational requirements. This is the only way to guarantee meaningful protection.

- Traffic Control: The first step is to control the data traffic, for example, through a firewall, which ideally not only separates the internal IT systems from the Internet but very precisely regulates which IT systems are allowed to communicate with which Operational technology (OT) systems, and also which protocols they are allowed to use for this. If, for example, an IT system should only communicate with an OT system via an HTTPS connection, it makes sense to limit communication to precisely this protocol [164–166]. This means that attacks based on the SMB (server message block) protocol, for example, are no longer possible.

- Conduct a risk assessment: The first step is to conduct a comprehensive risk assessment based on internal and external threats [167]. By doing so, specialists will understand their most vulnerable points and define security policies and risk migration [168].

- Design a security policy and processes: The cybersecurity policy of a power company provides a set of rules to follow. The purpose of an electric company's policy is to inform employees, suppliers, and other authorized users of their obligations concerning the protection of technological assets and information [169] and security policy violation [170]. One of the keys to maintaining a practical base is conducting a review once or twice a year.

- Execute projects that implement the risk mitigation plan: It is crucial to select a cybersecurity technology based on international standards [171,172].

- The anomaly detection by deep packet inspection, i.e., the "deep look" into the data communication, not only brings a considerable security advantage in the electrical industry but can also significantly increase productivity. In this way, new communication protocols, or even measured values that do not move within a defined framework, are recognized in real-time. This means that an attack or a creeping error can be reacted to very quickly before damage occurs. With this approach—after a learning phase—the normal behavior of the system is known. Anything that deviates from it in any way is recognized as an anomaly and triggers an alarm. The reasons for such a deviation can be varied, for example, a defective sensor, a new notebook belonging to a service employee, or an attack by a virus.

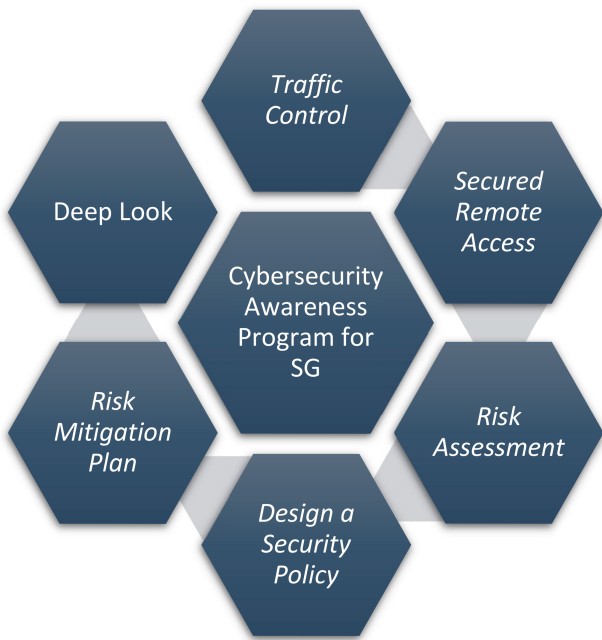

**Figure 4.** Mitigating the risk of cyber attack on smart grid systems.

## 7. Conclusions

The smart grid's basic idea is not enough when embarking on this complex system. Even with the available experiences and technologies, the ideal network's search is an investment based on time, money, and research. With the great efforts put forth for SG research, the power sector players pursue the energy revolution that humanity longs for.

The smart grid becomes more complex when environments involve numerous devices and increasing connectivity to other networks, including the Internet. For such systems, it is important to understand and comprehend the cyber elements and the implications of the integrated state of the environment. Furthermore, the diversity of the hardware and software in the SG sensors provides strong market competition, but this diversity is also a security issue in that there is no single security architect overseeing the entire "system" of the SG. Cybersecurity experts agree that standards alone will not provide the appropriate level of security.

Will artificial intelligence be the next step in our evolution? Although it is still in its infancy, AI is already changing the way we do things. Artificial intelligence, such as deep learning, are key topics that have been driving new technologies. AI technologies have great potential, especially when it comes to defending against cyber-attacks.

Despite existing guidelines and frameworks, designing and managing security for SG remains difficult. This paper identifies the trends, problems, and challenges of cybersecurity in smart grid critical infrastructures in big data and artificial intelligence. An extensive state-of-art analysis was completed—some specific guidelines for achieving cybersecurity awareness program for SG were discussed.

**Author Contributions:** Investigation, A.C.; writing—original draft preparation, A.C., I.F., X.Y.; writing—review and editing A.C., I.F., X.Y. All authors have read and agreed to the published version of the manuscript.

**Funding:** This research received no external funding.

**Conflicts of Interest:** The authors declare no conflict of interest.

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
