# Peer review of "Security Risk Modeling in Smart Grid Critical Infrastructures in the Era of Big Data and Artificial Intelligence"

_sustainability, doi:10.3390/su13063196_

Round 1

Reviewer 1 Report

This paper provides many references to the areas of smart grids, regarding aspects of cyber security attacks on their communication infrastructure. There is also a summary of papers regarding machine learning based approaches to increasing resilience against the attacks. However, the paper suffers from much general text which is not relevant to the subject of the title and abstract, there is no real analysis or discussion of the effectiveness or applicability of these approaches, no indication of key results or future directions for research or key questions that still need to be solved.

The authors have collected many references, and there are some good passages. Section 5.2 and Section 6 contain good introductory discussions, but tables 2 and 3, which seem to be the core results, are not further discussed and therefore do not achieve their full value. Section 6.2 contains a good summary of methods, but does not do more than re-word the titles of the cited papers. It does not achieve the goals set out at the start of the subsection (lines 315-319) – there is no evaluation, no discussion of what is appropriate or what shortcomings or possibilities for improvement of current plans there may be. Indeed there is no indication of the current state of the art on which to base this discussion.

Ironically, it seems that the recommendations made in Section 7 are standard cyber security techniques, and do not include any of the Big Data or AI techniques addressed in the paper.

To improve the paper, the following points (major and minor) should be addressed

  1. Overall there is much general text which is not interesting, or already known for experts, and which will not be sufficient to help a non-expert to understand the following discussion.
  2. Section 2 is very general and includes discussion of smart cities and mobility which is not relevant. Better would be a brief description (with references) of what the smart grid is, what security risk modeling is, how Big Data is becoming more relevant (if this is the case), and what the success of AI based approaches to date are.
  3. Section 3 is a general description of Security threats to IT systems. The link to smart grid infrastructure has not been made. The link: Smart Grid => SCADA System => IT Infrastructure needs to be established. It seems to be implicitly assumed that Smart Grids are centrally controlled, but there are also decentralized approaches. How does this impact the approach?
  4. Figure 1 is general & the applicability to Smart Grids not indicated. Delete or link to the topic.
  5. Figure 2 is a graphic of the text in lines 140 to 148 & adds no value.
  6. Table 1 is not smart grid specific.
  7. 4 again very general. Could be combined with Section 3, and then supported with references to other surveys.
  8. Section 5 Introduction, Figure 3 is generic to IT infrastructure. The implication is that the SG consists of Meters and Communication systems. But there are also the SCADA systems and substations which are ignored, but which are known to be the target of Cyber-Attacks. These topics need to be addressed.
  9. Section 5.1 is very generic & could be deleted without detracting from the paper. However there would then be nothing of “Big Data” left in the paper. The growing importance of Big Data specifically to SG should be established.
  10. Section 5.2 contains a good discussion. Table 2 is valuable. However there should be a deeper analysis of the state of the art and discussion of open questions and directions for future research. The methodology for the survey should be described (as in Section 6.1)
  11. Section 6 contains some good discussion, but the introductory discussion is general & should be shortened
  12. Section 6.1 has a good description of the methodology, and Table 3 is interesting. I do not understand why there is almost no overlap between Table 2 and Table 3.
  13. The CORAS Method is also described very well in [42], but this is not referenced in the description in Section 6.1. There are certainly other references.
  14. There is no discussion of the papers in Table 3. What is the quality of these papers? Which are particularly good? Are there commonalities? Which make a significant or original contribution? What topics are still open?
  15. Section 6.2 – This could be another table. The goals set out in lines 315 – 319 are not addressed. There is no analysis of the papers mentioned in the methods.
  16. Why is there no overlap between the references in Table 3 and the Methods in Section 6.2?
  17. Section 7 is good, but is standard (IT System) Cyber Security. What is specific / relevant to SG? What is a result of Big Data and AI aspects?
  18. Section 8 is missing
  19. There are survey papers addressing your topic, and you have copied from them, e.g. [25]. You only copy Fig 1, and connect it to Confidentiality issues, but it covers much more. Similarly, [42] has a deeper analysis of CORAS. What is the contribution of your paper with respect to these papers?

Author Response

Reviewer: 1

Thanks a lot for such motivational and encouraging comments. The suggestion provided by

the reviewer is really helpful in improving the overall quality of the manuscript.

To improve the paper, the following points (major and minor) should be addressed

Comment 1: Overall there is much general text which is not interesting, or already known for experts, and which will not be sufficient to help a non-expert to understand the following discussion.

Dear Reviewer, in the revised manuscript, the changes have been incorporated (we added new paragraph and we removed few sentences).

Comment 2: Section 2 is very general and includes discussion of smart cities and mobility which is not relevant. Better would be a brief description (with references) of what the smart grid is, what security risk modeling is, how Big Data is becoming more relevant (if this is the case), and what the success of AI based approaches to date are.

Dear Reviewer, thank you so much for the valuable and encouraging comments.

Section 2 deals with “Energy Management in Smart Sustainable Cities” we believe it important to add a section for Smart Sustainable Cities, since the topic of this journal and for this SI. We also believe that this section links Smart Sustainable Cities and Smart grid. Regarding the Smart grid, as per the review suggestion, we add incorporated the suggested changes.

Comment 3:  Figure 1 is general & the applicability to Smart Grids not indicated. Delete or link to the topic.

Dear Reviewer, in the revised manuscript, the changes have been incorporated. As per the reviewer's suggestion, we removed figure 1.

We have added the following paragraph

One of the sources of vulnerability resulting from integrating ICTs to SG is that all devices pass their data through the public network that is the Internet using the Internet Protocol (IP). However, this protocol has known weaknesses that can facilitate the risks of intrusions or interceptions of data. Yet, they have serious security gaps. Therefore, the safety in smart grids implies the protection and security of information.

Comment 4:  Figure 2 is a graphic of the text in lines 140 to 148 & adds no value

Dear Reviewer, in the revised manuscript, the changes have been incorporated. As per the reviewer's suggestion, we removed figure 2.

Comment 5:  Table 1 is not smart grid specific.

Dear Reviewer, we agree with your comment. However, we linked table 1 to the smart grid (the same was as for comment 3).

We have added the following paragraph

One of the sources of vulnerability resulting from integrating ICTs to SG is that all devices pass their data through the public network that is the Internet using the Internet Protocol (IP). However, this protocol has known weaknesses that can facilitate the risks of intrusions or interceptions of data. Yet, they have serious security gaps.

Comment 6:  Section 4 again very general. Could be combined with Section 3, and then supported with references to other surveys.

As per the reviewer's suggestion, we removed Section 4 and we combined it with Section 3. The changes have been made in the revised manuscript,

Comment 7:  Section 5 Introduction, Figure 3 is generic to IT infrastructure. The implication is that the SG consists of Meters and Communication systems. But there are also the SCADA systems and substations which are ignored, but which are known to be the target of Cyber-Attacks. These topics need to be addressed.

Dear Reviewer, in the revised manuscript, the changes have been incorporated.

We have added the following paragraph

The power grid's intelligence lies precisely in the process where multiple sensors convey precise, real-time information on the state of the grid to a system such as SCADA (supervisory control and data acquisition), which will capture, analyze, and react quickly accordingly.

While SCADA security is usually based on firewalls whose mandate allows access to authorized users only, they can't provide this protection if the cyber intrusion disguises itself as an identity authorized to impersonate legitimate.

Some SCADA systems were put in place dozens of years ago and are now impossible to update. They must then be replaced by more recent, safer equipment, but this is synonymous with significant investments and, therefore, often postponed.

Comment 8:  Section 5.1 is very generic & could be deleted without detracting from the paper. However there would then be nothing of “Big Data” left in the paper. The growing importance of Big Data specifically to SG should be established.

Dear Reviewer, in the revised manuscript, the changes have been incorporated.

We have added the following paragraph

Thanks to forward-looking algorithms (i.e., prediction of consumption according to the weather, forecasting of production, etc.), the SG has a global vision in real-time or in advance of these energy offers and demands.

The Smart Grid's strength lies in using this data to automatically adjust the energy flows of the network to supply areas of energy need with electricity primarily from renewable sources.

Electricity distributors are now actively engaged in a double movement towards Big Data - the quantitative explosion of data digital available - and to Open Data - the update free disposal of this data in an open manner, which allows their reuse without technical restriction.

Comment 9:  Section 5.2 contains a good discussion. Table 2 is valuable. However there should be a deeper analysis of the state of the art and discussion of open questions and directions for future research. The methodology for the survey should be described (as in Section 6.1)

Dear Reviewer, in the revised manuscript, the changes have been incorporated.

We have added the following paragraph

Cybercriminals have long since discovered the advantages of Artificial Intelligence to attack their victims in a more targeted, faster, and more sophisticated way.

When it comes to cybersecurity, protection against hackers, the electric utility companies are challenged more than ever before.  Organized cyber criminals who are well equipped both technically and financially are now in a position to develop highly intelligent attacks, which, thanks to capabilities such as complex obfuscation techniques, can hardly be detected by standard protection solutions.

Comment 10:  Section 6 contains some good discussion, but the introductory discussion is general & should be shortened

We would like to thank the reviewer for such productive and worthy suggestions.

As per the reviewer's suggestion, the introductory discussion of Section 6 (now Section 5) is shortened. The changes have been made in the revised manuscript.

Comment 11:  Section 6.1 has a good description of the methodology, and Table 3 is interesting. I do not understand why there is almost no overlap between Table 2 and Table 3.

Dear reviewer thank you a lot for such motivational and encouraging comments.

Table 2, related to subsection "4.2. Cybersecurity and Artificial Intelligence", summarized few works related to Artificial intelligence for a smarter kind of cybersecurity in the smart grid. The AI is changing the game for cybersecurity, analyzing massive quantities of risk data to speed response times and augment under-resourced security operations. While the Table 3, related to subsection "5.1. CORAS method for security risk analysis", focuses its intention on the CORAS technique. CORAS provides a customized language for threat and risk modeling and comes with detailed guidelines explaining how the language should be used to capture and model the relevant smart grid's information during the various security analysis stages.

Both topics attract many scientists from both academic and industrial environments. Artificial intelligence (AI) and CORAS are playing an increasing role in cybersecurity, with security tools analyzing data.

It is worth to mention that the Table 2 and Table 3 shared some references (i.e., [47], [48], [49], [52], [57], [64],[67], [68], [69]).

Comment 12:  The CORAS Method is also described very well in [42], but this is not referenced in the description in Section 6.1. There are certainly other references.

Dear Reviewer, thanks for the suggestion; in the revised manuscript, the suggestion is

incorporated.

Comment 13:    There is no discussion of the papers in Table 3. What is the quality of these papers? Which are particularly good? Are there commonalities? Which make a significant or original contribution? What topics are still open?

Dear Reviewer, Table 3 contains over 100 references. Those papers have all of a good/excellent quality. Those works are peer-reviewed articles collected from IEEEXplore and SpringerLink databases. Due to space limitations, we only provided references without discussing each paper separately.

Comment 14:  Section 6.2 – This could be another table. The goals set out in lines 315 – 319 are not addressed. There is no analysis of the papers mentioned in the methods.

We would like to thank the reviewer for such productive and worthy suggestions.

as per the review suggestion, we add new table (i.e. Table 4).

Dear Reviewer, due to space limitations, we only discussed the summary of each method. For each method we cited the main reference.

Comment 15:  Why is there no overlap between the references in Table 3 and the Methods in Section 6.2?

We would like to thank the reviewer for your suggestions. To avoid making the text more cumbersome, we have decided to separate the SG risk and the list of the risk assessment methods for SCADA systems. As per your suggestion, we create another table with the most effective risk assessment methods (i.e. Table 4).

Comment 16:  Section 7 is good, but is standard (IT System) Cyber Security. What is specific / relevant to SG? What is a result of Big Data and AI aspects?

Dear reviewer thanks you a lot for such motivational and encouraging comments.

In the revised manuscript, the changes have been incorporated. We have added the following paragraph

The traditional tiered approach to cybersecurity can only prevent and detect the less elaborate threats. In the meantime, modern cyber-attacks are carefully designed to bypass standard security controls by learning detection rules. In addition, traditional controls may not adequately counter insider threats, a form of insidious attack launched by those with legitimate access.

By leveraging AI and advanced big data analytics, cybersecurity technologies can generate predictive and actionable insights that will help you make better cybersecurity decisions and protect your smart grid against threats. They can also help the electric utility detect and counter threats faster by monitoring the cyber environment at speed and with a precision level that only machines can.

Artificial intelligence technologies are already integrated into tools such as antivirus, EDR (Endpoint Detection and Response) solutions, firewalls, Data Loss Prevention, etc., that automatically respond to attacks by filtering malicious traffic. Vulnerability management has become a point of tension for operational teams due to the constant increase in the number of known vulnerabilities, difficulties in assessing the real risks induced and prioritizing and automating patches' deployment. Indeed, of the thousands of vulnerabilities published each year, only a fraction is used by attackers. Besides, some systems are protected by perimeter defenses.

This complexity is driving vulnerability management tool vendors to integrate AI technologies into their solutions. The objective of AI applied to vulnerability management is to improve the discovery of active equipment, the scanning of vulnerabilities, the determination of associated risks connected with intelligence on the threat, the prioritization, and deployment of patches.

Comment 17:  Section 8 is missing

Dear Reviewer, thanks for highlighting our mistake; the section number have been updated.

Comment 18:   There are survey papers addressing your topic, and you have copied from them, e.g. [25]. You only copy Fig 1, and connect it to Confidentiality issues, but it covers much more. Similarly, [42] has a deeper analysis of CORAS. What is the contribution of your paper with respect to these papers?

We referenced and used both works. Unlike the work [25] and [42], in this paper, we covered several topics and challenges related to cybersecurity in the smart grid network.  An extensive state-of-art was completed. More than 179 papers were reviewed and cited, several of these papers were published in 2019 and 2020.

We also incorporate AI and Big data analytics in smart grid security. These days, AI technologies are generating a lot of excitement. As we said previously, AI is changing the game for cybersecurity. Now is the time to walk the talk. Here are several steps to take to enhance the cyber capabilities of the smart grid through the use of AI and big data analytics technologies.

We also provide state of the art on cybersecurity risk assessment methods for SCADA systems. We also provided a review and summarized several techniques on how to mitigate the risk of the cyber attack on smart grid systems.

Reviewer 2 Report

Abstract:

We review exposure levels while proposing suitable security countermeasures. Finally, smart grid's cyber-security risk assessment methods for supervisory control and data acquisition are presented.

Line 48-49: As the electrical grid merges and becomes "smarter" with the resultant benefits of a better connectivity, cybersecurity risks and threats also increase.

Introduction:

good, no issues

Lines 96-98: The implementation of the smart and sustainable city, a complex system, requires new governance involving all the connected actors - local communities, companies, citizens - and a lot of research is required to draw its contours.

2. Energy Management in Smart Sustainable Cities

Lines 114-115: Another major challenge awaits urban agglomerations: according to a recent WHO report [NEEDS IN-TEXT NUMERICAL REFERENCING HERE], 92% of urban populations do not breathe healthy air.

Lines 130-131: ...g is, when offices are empty - to homes. Electric vehicles can be called upon to provide electricity during ...

3. Security Threats in Smart Grids

Table 1 - left hand column>

Modification (when accessing) and modifications are made to
data, environmental devices, or components (cyber deliberately and illegally)

4. Security Requirements in Smart Grids

no issues - fine

5. Security-Aware of SG Infrastructures in Era of Big Data and Artificial Intelligence

Line 218-219: The most important resource in the world is no longer crude oil, but data - according to The Economist's title from ...

Table 2: Cybersecurity without Errors

Lines 271-272: When it comes to cybersecurity, protection against hackers, REMOVE "THE" electric utility companies are challenged more than ever before

Line 296: A literature review is used in this article to explore various security modeling techniques ...

7. Mitigating the Risk of Cyber Attack on Smart Grid Systems

Line 354-355: Electricity companies must implement a complete program that integrates a good organization and adequate processes.

8. Conclusion   THE NUMBERING IN THE TEXT = 9 = IS WRONG!

ok otherwise

References

be consistent in presentation - year of publication always at the end

Author Response

Security Risk Modeling in Smart Grid Critical Infrastructures in the Era of Big Data and Artificial Intelligence

Answers to reviewers' comments

Colour Schemes:

Black: Original comment.

Blue: For Response to Reviewer

Dear Editor and Reviewers,

We would like to thank you for the careful and thorough reading of this manuscript and

the thoughtful and supportive comments and constructive suggestions, which help

improve this manuscript's quality. Please see below, in blue, our detailed response to

comments.

Reviewer: 2

We review exposure levels while proposing suitable security countermeasures. Finally, smart grid's cyber-security risk assessment methods for supervisory control and data acquisition are presented.

Thanks a lot for such motivational and encouraging comments. The suggestion provided by

the reviewer is really helpful in improving the overall quality of the manuscript.

Comment 1: Line 48-49: As the electrical grid merges and becomes "smarter" with the resultant benefits of a better connectivity, cybersecurity risks and threats also increase.

Dear Reviewer, thank you for the valuable suggestion. In the revised manuscript, the changes have been incorporated.

Comment 2: Introduction: good, no issues

Thanks a lot for such encouraging comments.

Comment 3: Lines 96-98: The implementation of the smart and sustainable city, a complex system, requires new governance involving all the connected actors - local communities, companies, citizens - and a lot of research is required to draw its contours.

Dear Reviewer, thank you for the valuable suggestion. In the revised manuscript, the changes have been incorporated.

Comment 3: 2. Energy Management in Smart Sustainable Cities

Comment 4: Lines 114-115: Another major challenge awaits urban agglomerations: according to a recent WHO report [NEEDS IN-TEXT NUMERICAL REFERENCING HERE], 92% of urban populations do not breathe healthy air.

Dear Reviewer, thank you for highlighting our mistake. In the revised manuscript, the changes have been incorporated.

Comment 5: Lines 130-131: ...g is, when offices are empty - to homes. Electric vehicles can be called upon to provide electricity during ...

Dear Reviewer, thank you for the valuable suggestion. In the revised manuscript, the changes have been incorporated.

Comment 6: 3. Security Threats in Smart Grids

Comment 7: Table 1 - left hand column>

Modification (when accessing) and modifications are made to
data, environmental devices, or components (cyber deliberately and illegally)

Dear Reviewer, thank you for highlighting our mistake. In the revised manuscript, the changes have been incorporated.

Comment 8: 4. Security Requirements in Smart Grids

no issues – fine

Thanks a lot for such encouraging comments.

Comment 9: 5. Security-Aware of SG Infrastructures in Era of Big Data and Artificial Intelligence

Comment 10: Line 218-219: The most important resource in the world is no longer crude oil, but data - according to The Economist's title from ...

Dear Reviewer, thank you for highlighting our mistake. In the revised manuscript, the changes have been incorporated.

Comment 11: Table 2: Cybersecurity without Errors

Dear Reviewer, thank you for highlighting our mistake. In the revised manuscript, the changes have been incorporated.

Comment 12: Lines 271-272: When it comes to cybersecurity, protection against hackers, REMOVE "THE" electric utility companies are challenged more than ever before

Dear Reviewer, thank you for highlighting our mistake. In the revised manuscript, the changes have been incorporated.

Comment 13: Line 296: A literature review is used in this article to explore various security modeling techniques ...

Dear Reviewer, thank you for highlighting our mistake. In the revised manuscript, the changes have been incorporated.

Comment 14: 7. Mitigating the Risk of Cyber Attack on Smart Grid Systems

Line 354-355: Electricity companies must implement a complete program that integrates a good organization and adequate processes.

We would like to thank the reviewer for such productive and worthy suggestions. In the revised manuscript, the changes have been incorporated.

Comment 15: 8. Conclusion   THE NUMBERING IN THE TEXT = 9 = IS WRONG!

Dear Reviewer, thanks for highlighting our mistake; the section number have been updated.

Comment 16: ok otherwise

Thanks a lot for such encouraging comments.

Comment 17: References

be consistent in presentation - year of publication always at the end

We would like to thank the reviewer for such productive and worthy suggestions.  As per the review suggestion, all the reference list was updated with the year of publication always at the end.

Reviewer 3 Report

The paper is interesting, good, innovative and also well written.

It lacks a clear overview and the description of the structure of the paper at the end of the introduction.

Some improvements are required before paper acceptance. In particular, a revision of the abstract is required since the second part [preparing...... road map] is completely unreadable and could mislead the reader on the real objective of this paper.

I think that this paper requires a few improvements before the final acceptance as below:

  1. First of all the title of this paper needs to get improved based on your article framework
  2. The abstract needs to get improved and in this case the first part of the abstract is actually a part that could be included in the paragraph of Introduction.
  3. It is necessary to highlight the novelty and contribution of the paper in the introduction.
  4. It is necessary to add some information about methodology and key results to make them clear.
  5. The introduction needs to improve, I am thinking about lacking theoretical background of the study. This should be located in the Introduction or in independent section.
  6. I suggest to the authors to add a new part named result and discussion 
  7. The conclusion part must be improved based on the academic paper.
  8. That would be good if you references also previous studies in the field.
  9. The results should be gathered independently so that authors‘ contribution is clear.
  10. Please follow standardized structure of academic papers (Introduction, Background, Theory, Methodology, Case studies, Results, Discussion, Conclusions and Limitations and Future Research). It makes orientation in the paper for international reader easier.
  11. some of your figures are not visible like Figure 1 and 4

Author Response

Security Risk Modeling in Smart Grid Critical Infrastructures in the Era of Big Data and Artificial Intelligence

Answers to reviewers' comments

Colour Schemes:

Black: Original comment.

Blue: For Response to Reviewer

Dear Editor and Reviewers,

We would like to thank you for the careful and thorough reading of this manuscript and

the thoughtful and supportive comments and constructive suggestions, which help

improve this manuscript's quality. Please see below, in blue, our detailed response to

comments.

Reviewer: 3

The paper is interesting, good, innovative and also well written.

Thanks a lot for such motivational and encouraging comments.

It lacks a clear overview and the description of the structure of the paper at the end of the introduction.

Some improvements are required before paper acceptance. In particular, a revision of the abstract is required since the second part [preparing...... road map] is completely unreadable and could mislead the reader on the real objective of this paper.

Thanks a lot for such motivational and encouraging comments. The suggestion provided by

the reviewer is really helpful in improving the overall quality of the manuscript.

I think that this paper requires a few improvements before the final acceptance as below:

Comment 1: First of all the title of this paper needs to get improved based on your article framework

Dear Reviewer, in this work, we considered those data set which are already. We also provide state of the art on cybersecurity risk assessment methods. By leveraging AI and advanced big data analytics, cybersecurity technologies can generate predictive and actionable insights that will help you make better cybersecurity decisions and protect your smart grid against threats.

Comment 1: The abstract needs to get improved and in this case the first part of the abstract is actually a part that could be included in the paragraph of Introduction.

Dear Reviewer, thank you for the valuable suggestion. In the revised manuscript, the changes have been incorporated.

Comment 1: It is necessary to highlight the novelty and contribution of the paper in the introduction.

Dear Reviewer, thank you for the valuable suggestion. In this work, we conduct comprehensive overview and analysis of smart grid architecture and different security aspects in the era of big data and artificial intelligence.

Comment 1: It is necessary to add some information about methodology and key results to make them clear.

Dear Reviewer, in this work the electronic databases IEEEXplore and SpringerLink were used in this literature review. The work consists of a main qualitative study supplemented by a quantitative study.  Building a database isn't easy at it sounds, to create the comparative databases, search keywords used. Among these keywords, we can quote "attack tree security", "Vulnerability Analysis", "False Data Injection Attack Detection", "malicious behavior detection", "Deep Learning Detection of Electricity Theft Cyber-Attacks", "Fraud Detection", "bow tie security", "Anomaly Detection Method", "Smart Grids Cyber-Attack Defense", and "CORAS security". 

Comment 1: The introduction needs to improve, I am thinking about lacking theoretical background of the study. This should be located in the Introduction or in independent section.

Dear Reviewer, thank you for the valuable suggestion. In the revised manuscript, the changes have been incorporated.

Comment 1: I suggest to the authors to add a new part named result and discussion 

We would like to thank the reviewer for your suggestions. To avoid making the text more cumbersome, we have decided to discuss the SG risk and the list of the risk assessment methods in each section (rather than adding a whole new section). As per your suggestion, we add more discussion on the effective risk assessment methods (i.e., Table 4).

Comment 1: The conclusion part must be improved based on the academic paper.

Dear Reviewer, thank you for the valuable suggestion. In the revised manuscript, the changes have been incorporated.

Comment 1: That would be good if you references also previous studies in the field.

Dear Reviewer, thank you for the valuable suggestion. In this paper, we covered several topics and challenges related to cybersecurity in the smart grid network.  An extensive state-of-art was completed. More than 179 papers were reviewed and cited, several of these papers were published in 2019 and 2020.

Comment 1: The results should be gathered independently so that authors‘ contribution is clear.

Dear Reviewer, in the revised manuscript, the changes have been incorporated

Comment 1: Please follow standardized structure of academic papers (Introduction, Background, Theory, Methodology, Case studies, Results, Discussion, Conclusions and Limitations and Future Research). It makes orientation in the paper for international reader easier.

Dear Reviewer, thank you for the valuable suggestion. In this paper, a literature review is a research method used to address all relevant cybersecurity studies for the smart grid.  We also incorporate AI and Big data analytics in smart grid security. These days, AI technologies are generating a lot of excitement. As we said previously, AI is changing the game for cybersecurity. Now is the time to walk the talk. Here are several steps to take to enhance the cyber capabilities of the smart grid through the use of AI and big data analytics technologies.

We also provide state of the art on cybersecurity risk assessment methods for SCADA systems. We also provided a review and summarized several techniques on how to mitigate the risk of the cyber attack on smart grid systems.

Comment 1: some of your figures are not visible like Figure 1 and 4

Dear Reviewer, in the revised manuscript, the changes have been incorporated. As per the reviewer's suggestion, we removed figure 1.

Round 2

Reviewer 1 Report

Congratulations to the authors on this much improved version of the paper.  Thank you also for attending to my many comments. In some cases we may have to agree to disagree - e.g. I would prefer to see some more concrete statements on the future research directions.  However the paper has value in the great number of references and the recommendations in Section 6.

I have the following comments.

  1. lines 139-140.  I do not believe that this statement is correct. It seems to contradict the information in Fig 1.  Or perhaps I am interpreting "ICTs" incorrectly.  Many sensors used by utilities use other protocols than IP. Also Utilities often use private networks or VPN networks.  For an Overview, see for example "A survey on communication networks for electric system automation" by Güngor & Lambert, in particular Fig 2.
  2. Thank you for including SCADA in lines 206-214.  You make some broad statements here. Are there references to back this up? Lines 209-211 bring up a very important point, but the sentence is incomplete. Please re-word. Again here a reference would be good.  (minor comment)
  3. Section 6 is much improved with the new text. Again references - as in the existing text after line 388 would be appreciated. I assume these would be those of Table 4.
  4. Conclusion, First sentence.  I do not believe this is correct. Many devices installed in modern utility networks - phase changers, power electronic devices, saftey devices, SCADA systems at all network levels - did not exist at the end of the 19th century.  Indeed if there were no modern control devices, there would be no need for cybersecurity.  Please revise this first paragraph. What is the key message?
  5. Conclusion: You have made good contributions with your surveys & the great number of references & summaries & the recommendations in Section 6. You should mention these in the Conclusion.  Lines 433-436 are then a good wrap-up.
  6. The final sentence lines 437-438 is incomplete & needs revision, completion or deletion.

Author Response

Security Risk Modeling in Smart Grid Critical Infrastructures in the Era of Big Data and Artificial Intelligence

Answers to reviewers' comments

Color Schemes:

Black: Original comment.

Blue: For Response to Reviewer

Dear Editor,

We would like to thank you and all the reviewers for very constructive comments and useful

suggestions, which are truly helpful in improving our manuscript. We have studied carefully all the inputs and comments by your editorial board and revised the manuscript by taking into account all the suggestions/comments made by your reviewers.

Also, thank you for allowing a re-submission of our manuscript, with an opportunity to address the reviewers’ comments.

The detailed response, listing the actions we have taken where changes were requested, is, as you

suggested are given in the following sections below.

We are uploading

(a) our point-by-point response to the comments (below) (response to reviewers),

(b) an updated manuscript with yellow highlighting indicating changes.

For your and reviewer’s convenience, our responses are shown in blue color.

Hope you and the reviewers find the revision acceptable. Looking forward to hearing from you the final decision on our submission.

Regards,

Reviewer: 1

Congratulations to the authors on this much improved version of the paper.  Thank you also for attending to my many comments. In some cases we may have to agree to disagree - e.g. I would prefer to see some more concrete statements on the future research directions.  However the paper has value in the great number of references and the recommendations in Section 6.

Thanks a lot for such motivational and encouraging comments. The suggestion provided by

the reviewer is really helpful in improving the overall quality of the manuscript.

I have the following comments.

Comment 1: lines 139-140.  I do not believe that this statement is correct. It seems to contradict the information in Fig 1.  Or perhaps I am interpreting "ICTs" incorrectly.  Many sensors used by utilities use other protocols than IP. Also Utilities often use private networks or VPN networks.  For an Overview, see for example "A survey on communication networks for electric system automation" by Güngor & Lambert, in particular Fig 2.

Dear Reviewer, you are right and this is really very important observation.

As shown in Figre.1 there are various protocols and a myriad of information and telecommunications technologies that can operate in an electrical distribution system. However, as we have stated in lines 139-140, "One of the sources of vulnerability resulting from integrating ICTs to SG is that all devices pass their data through the public network that is the Internet using the Internet Protocol (IP)."

In fact, the IP protocol is one of my protocols that can be used in SG and hence one of many source of SG vulnerability. Sensors and actuators, and other ICT devices have been used in the current grid for years. Their number will dramatically increase over the next few years in all Smart Grid areas, from distribution to homes and buildings. Their communication models will be based on IP (or IPv6), end to end, which undoubtedly meets all of Smart Grid networks' stringent requirements. One reason why IP has only recently won careful consideration in the smart grid network infrastructure is a misconception that the protocol was a "heavyweight," requiring significant computing power and memory to run in the constrained environment of metering infrastructure.  These IP-enabled smart objects connected to multi-service IP networks are one major component of Smart Grid networks.

Comment 2: Thank you for including SCADA in lines 206-214.  You make some broad statements here. Are there references to back this up? Lines 209-211 bring up a very important point, but the sentence is incomplete. Please re-word. Again here a reference would be good.  (minor comment)

Dear Reviewer, thanks for the suggestion; in the revised manuscript, the suggestion is incorporated. Kindly refer Section 3 on page 6.

In addition, we added following paragraph

Some SCADA systems or elements were put in place dozens of years ago and are now impossible to update. Some of them were designed before well-founded cybersecurity principles were settled upon. SCADA system designers would claim that cybersecurity is not a concern since SCADA systems are not connected to the Internet. However, over time, SCADA systems began appearing on the Internet, and often with no cybersecurity. These systems must be replaced by more recent, safer equipment, but this is synonymous with significant investments and, therefore, often postponed.

On larger sites, the control system needs to be protected from attack within the SCADA network. Implementing an additional firewall between the corporate and SCADA network can achieve by imposing more restrictive rules. This will enable authorized service engineers to provide support and manage security, e.g., apply security mitigations, inspect log files, apply updates, etc.

Comment 3: Section 6 is much improved with the new text. Again references - as in the existing text after line 388 would be appreciated. I assume these would be those of Table 4.

Dear reviewer, thank you very much for such encouraging remark.

Yes, you are right. The references were summarized in Table 4 (as you have suggested previously).

Comment 4: Conclusion, First sentence.  I do not believe this is correct. Many devices installed in modern utility networks - phase changers, power electronic devices, saftey devices, SCADA systems at all network levels - did not exist at the end of the 19th century.  Indeed if there were no modern control devices, there would be no need for cybersecurity.  Please revise this first paragraph. What is the key message?

Dear Reviewer, the changes have been made in the revised manuscript.  The first sentence of Conclusion has been deleted.

Comment 4: Conclusion: You have made good contributions with your surveys & the great number of references & summaries & the recommendations in Section 6. You should mention these in the Conclusion.  Lines 433-436 are then a good wrap-up.

Thanks a lot for such motivational and encouraging comments. The suggestion provided by

the reviewer is really helpful in improving the overall quality of the manuscript. In the revised manuscript, the changes have been made.

Furthermore, we added following paragraph

The smart grid becomes more complex environments involve numerous devices and increasing connectivity to other networks, including the Internet. For such systems, it is important to understand and comprehend the cyber elements and the implications of the integrated state of the environment. Furthermore, the diversity of the hardware and software in the SG sensors provides strong market competition, but this diversity is also a security issue in that there is no single security architect overseeing the entire "system" of the SG.

Will artificial intelligence be the next step in our evolution? Although it is still in its infancy, AI is already changing the way we do things. Artificial intelligence, such as deep learning, are key topics that have been driving new technologies. AI technologies have great potential, especially when it comes to defending against cyber-attacks.

Despite existing guidelines and frameworks, designing and managing security for SG remains difficult. This paper identifies the trends, problems, and challenges of cybersecurity in smart grid critical infrastructures in big data and artificial intelligence.  An extensive state-of-art was completed—some specific guidelines for achieving cybersecurity awareness program for SG we discussed.

Comment 5: The final sentence lines 437-438 is incomplete & needs revision, completion or deletion.

Dear Reviewer, the changes have been made in the revised manuscript. The final sentence lines 437-438 have been deleted.
